# The ultrastructure of infectious L-type bovine spongiform encephalopathy prions constrains molecular models

Razieh Kamali-Jamil[1,2], Ester Vázquez-Fernández[1,2], Brian Tancowny[1,2], Vineet Rathod[1,2], Sara Amidian[1,2], Xiongyao Wang[1,2¤], Xinli Tang[1,2], Andrew Fang[1,2], Assunta Senatore[3], Simone Hornemann[3], Sandor Dudas[4], Adriano Aguzzi[3], Howard S. Young[1], Holger Wille[1,2,5]*

1 Department of Biochemistry, University of Alberta, Edmonton, Alberta, Canada, 2 Centre for Prions and Protein Folding Diseases, University of Alberta, Edmonton, Alberta, Canada, 3 Institute of Neuropathology, University of Zürich, Zürich, Switzerland, 4 Canadian BSE Reference Laboratory, Canadian Food Inspection Agency, Lethbridge Laboratory, Lethbridge, Alberta, Canada, 5 Neuroscience and Mental Health Institute, University of Alberta, Edmonton, Alberta, Canada

¤ Current address: School of Materials Science and Engineering, Harbin Institute of Technology, Weihai, Shandong, China
* wille@ualberta.ca

**Data Availability Statement:** Most relevant data are contained within the paper and its Supporting Information file. Additional raw data (electron

## Abstract

Bovine spongiform encephalopathy (BSE) is a prion disease of cattle that is caused by the misfolding of the cellular prion protein (PrP$^C$) into an infectious conformation (PrP$^{Sc}$). PrP$^C$ is a predominantly α-helical membrane protein that misfolds into a β-sheet rich, infectious state, which has a high propensity to self-assemble into amyloid fibrils. Three strains of BSE prions can cause prion disease in cattle, including classical BSE (C-type) and two atypical strains, named L-type and H-type BSE. To date, there is no detailed information available about the structure of any of the infectious BSE prion strains. In this study, we purified L-type BSE prions from transgenic mouse brains and investigated their biochemical and ultra-structural characteristics using electron microscopy, image processing, and immunogold labeling techniques. By using phosphotungstate anions (PTA) to precipitate PrP$^{Sc}$ combined with sucrose gradient centrifugation, a high yield of proteinase K-resistant BSE amyloid fibrils was obtained. A morphological examination using electron microscopy, two-dimensional class averages, and three-dimensional reconstructions revealed two structural classes of L-type BSE amyloid fibrils; fibrils that consisted of two protofilaments with a central gap and an average width of 22.5 nm and one-protofilament fibrils that were 10.6 nm wide. The one-protofilament fibrils were found to be more abundant compared to the thicker two-protofilament fibrils. Both fibrillar assemblies were successfully decorated with mono-clonal antibodies against N- and C-terminal epitopes of PrP using immunogold-labeling techniques, confirming the presence of polypeptides that span residues 100–110 to 227–237. The fact that the one-protofilament fibrils contain both N- and C-terminal PrP epitopes constrains molecular models for the structure of the infectious conformer in favour of a compact four-rung β-solenoid fold.

micrographs) and 3D reconstructions are deposited in figshare: https://doi.org/10.6084/m9.figshare.14600310.v1; https://doi.org/10.6084/m9.figshare.14600307.v1.

**Funding:** Funding for this study was provided by the Alberta Prion Research Institute / Alberta Innovates (awards 201300012 and 201600029 to HW) and the Canadian Institutes of Health Research (award ERL 138396 to HW). The funders had no role in study design, data collection and analysis, decision to publish, or preparation of the manuscript.

**Competing interests:** The authors have declared that no competing interests exist.

## Author summary

Bovine spongiform encephalopathy (BSE), also called "mad cow disease," is a deadly neurodegenerative disease in cattle. BSE is caused by $PrP^{Sc}$, which is an aberrantly folded conformer of a normal protein in the host. $PrP^{Sc}$ is an infectious protein and also referred to as a prion. BSE prions exist in three variants or strains: C-type BSE prions, which caused the epizootic "mad cow disease" outbreak, and two atypical forms L-type and H-type BSE prions, named according to their migration patterns during gel electrophoresis. For our investigations, we isolated L-type BSE prions from transgenic mouse brains and analyzed these samples using transmission electron microscopy and three-dimensional reconstruction techniques. Our study revealed that L-type BSE prions assemble into one- and two-protofilament containing amyloid fibrils and that the width of the two-protofilament fibrils is approximately twice that of one-protofilament fibrils. In addition, we labeled the L-type BSE fibrils at the ultrastructural level using specific anti-prion protein antibodies that recognize epitopes at both ends of the molecule. Our data agree with the previously proposed four-rung β-solenoid model for the structure of infectious $PrP^{Sc}$.

## Introduction

Prion diseases, also called transmissible spongiform encephalopathies (TSE), belong to a group of zoonotic, fatal neurodegenerative diseases that cause spongiform changes in the brain [1]. Bovine spongiform encephalopathy (BSE), also known as 'mad cow disease', was first detected in 1986 in the U.K. [2], and later was identified in other European countries and North America [3]. So far, BSE is the only zoonotic prion disease with the confirmed capability of transmission to humans, resulting in the occurrence of variant Creutzfeldt-Jakob disease, an acquired form of prion disease in humans. Thus, BSE was considered an essential human health risk [4–6]. Three BSE strains have been reported to cause prion disease in cattle, including classical BSE (C-type) and two atypical forms termed L-type and H-type BSE, which differ in their neuropathological and molecular phenotypes [7–9]. The disease caused by L-type BSE is also called bovine amyloidotic spongiform encephalopathy (BASE), due to the atypical deposition of amyloid plaques in the brain [8,10].

The underlying pathogenic event for all prion diseases involves the conversion from the normal, cellular prion protein, $PrP^C$, to an infectious form, known as $PrP^{Sc}$ [11]. According to previous nuclear magnetic resonance (NMR) spectroscopic and X-ray crystallographic studies on recombinant PrP, the $PrP^C$ structure contains an unstructured N-terminal domain and a C-terminal domain containing three α-helices and two short, antiparallel β-strands [12–14]. The unstructured N-terminal domain of bovine $PrP^C$ contains five or six octapeptide-repeats, instead of the five repeats that are found in most mammalian species [15]. In contrast to $PrP^C$, mammalian $PrP^{Sc}$ is difficult to study at the structural level due to its insolubility and propensity to aggregate [16]. Nevertheless, data obtained from a combination of structural experiments, including Fourier-transform infrared spectroscopy (FTIR), limited proteolysis, X-ray fiber diffraction, cryo electron microscopy (cryo-EM), molecular dynamics simulations, and solid-state NMR spectroscopy (ssNMR), have provided basic insights into the unique arrangements of this incompletely understood conformation [17,18].

Earlier structural studies reported that $PrP^{Sc}$ has a high β-sheet content and suggested that it retained part of the α-helices that are present in cellular $PrP^C$ [19–21]. However, a hydrogen/deuterium (H/D) exchange study demonstrated that the $PrP^{Sc}$ structure is likely devoid of

α-helices and predominantly contains β-strands [22]. Later, a four-rung β-solenoid model (4RβS) was proposed for the core structure of the infectious prion protein based on an X-ray fiber diffraction study on amyloid fibrils from N-terminally truncated PrP^Sc, termed PrP 27–30 [23]. The 4RβS arrangement was further supported by a cryo-EM analysis and three-dimensional reconstruction experiments on brain-derived and GPI-anchorless mouse prion fibrils [24]. Recently, an atomistic model for mouse PrP^Sc was created based on the 4RβS arrangement and employing results from the previous cryo-EM and X-ray fiber diffraction studies, as well as all other experimental data. Molecular Dynamics (MD) simulations were used to test this model, which was found to be physically stable [25]. Results of a recent ssNMR study on infectious recombinant PrP^Sc were found to be compatible with the four-rung β-solenoid model [26]. A parallel in-register intermolecular β-sheet (PIRIBS) structure was proposed as an alternative model for the architectures of PrP^Sc and recombinant PrP amyloid [27]. Recently, a high-resolution cryo-EM study of brain-derived 263K prions revealed a PIRIBS structure as the underlying fold for this prion strain [28]. According to this structure, monomers of β-structured prion protein stack on top of each other in-register with a 4.8 Å height increase per molecule, requiring the whole cross-section of the fibril to accommodate the full length of the peptide chain [28,29].

While there have been a variety of structural experiments on recombinant and tissue-derived scrapie prion strains, very little is known about the structural characteristics of native BSE prions. It has been previously established that different prion strains may hold variations in their structural conformation and self-replication procedures [4,30]. In this regard, we aimed at identifying the structural features of brain-derived infectious BovPrP^Sc by purifying BSE prions from transgenic mouse brains and employing electron microscopic approaches, such as image processing and immunogold labeling. For this purpose, we selected L-type BSE prions, as this strain has been demonstrated to be more amyloidogenic compared to the classical and H-type strains [8]; therefore, our first goal was to isolate L-type BSE prion fibrils for structural analyses. Here, we provide a first, detailed description of infectious, fully glycosylated, and GPI-anchored BSE prion fibrils. Our study revealed that the infectious BSE prion monomers polymerize into unbranched, long, rod-like fibrils with one- and two-protofilament morphologies, with the former type being significantly more prevalent. This finding is inconsistent with the stacked β-sheet model, as, in this configuration, the fold requires the width of a double-wide amyloid fibril. Therefore, the one-protofilament L-type BSE amyloid fibrils provide further support for the compact, four-rung β-solenoid model for the structure of PrP^Sc.

## Results

### Propagation of L-type BSE prions in transgenic mice expressing bovine PrP

Canada has detected a total of 19 BSE field cases since May 2003; one of them was typed as an L-type, atypical BSE case [10]. This animal was a Hereford pure breed heifer that was 13.7 years old when it was euthanized and tested positive for L-type BSE. Brain samples from this case were confirmed BSE positive at the Canadian National BSE Reference Laboratory. Western blot typing identified that the BSE prions in the brain of this heifer were consistent with L-type BSE. As a field case animal, the tissue condition was poor, and the amount available was limited. In order to generate more good quality tissue for follow up studies, brain tissue from the Canadian L-type BSE field case was passaged into 2 calves. Brain homogenate samples from these L-type BSE affected cows were used for intracerebral inoculation of transgenic mice overexpressing bovine PrP (Tg4092) [31]. Brains from terminally sick Tg4092 mice were used for the purification and subsequent ultrastructural characterization of L-type BSE prions.

## Isolation of L-type BSE prions

Digestion with proteinase K (PK) has been employed as a standard approach to diagnose prion formation for many years [1,32,33], and it has been shown that proteolysis with PK does not alter the self-replication and disease-causing capability of the resulting PK-digested prions [34,35]. Here, we used the PK digestion method to eliminate PrP$^C$ and other proteins from the purified samples. The purification procedure also included PTA, as this agent has been demonstrated to facilitate selective precipitation of PrP$^{Sc}$ and PrP 27–30 compared to PrP$^C$ [36,37]. Following the isolation of L-type BSE prions using PK and PTA, the quality and purity of the samples were analyzed using Western blots and silver-stained SDS gels. The latter showed signals in the pellet fraction while no protein was detected in the supernatant fraction, suggesting an appropriate purity of the samples (Fig 1A). Western blots of samples collected during the purification indicated the stepwise enrichment of the prion protein. The final pellet showed the typical three bands corresponding to non-, mono-, and diglycosylated PrP 27–30 when detected using the anti-PrP monoclonal antibody D15.15 (Fig 1B).

We also implemented a modified purification procedure, including a sucrose step-gradient centrifugation to remove extraneous lipids and peptides, which interfere with the ultrastructural analysis using electron microscopy. For this purpose, the semi-purified P1 pellet from the PTA purification was loaded onto a layered sucrose gradient cushion (40% and 80% sucrose), which separates the prion protein from the lipids, peptides, and PTA according to their densities. The prion protein with a density of approximately 1.2 g/ml was expected to migrate to the interface between 40% and 80% sucrose, with 1.1 g/ml and 1.4 g/ml solution densities, respectively. After ultracentrifugation, fractions were collected from the top and examined by immunoblotting and silver stain SDS gels (Fig 1C and 1D). As predicted, Western blots showed strong PK-resistant PrP 27–30 signals in the middle fraction compared to the fractions from higher and lower density areas. The top fractions and the bottom fractions showed relatively weak PrP 27–30 signals only. On the other hand, the pellet-wash fraction that was recovered once all fractions were collected, contained a high intensity PrP 27–30 signal (Fig 1D), which can be explained by the pelleting of prion aggregates complexed with PTA, resulting in a particle density in excess of 1.4 g/ml [38].

In a separate Western blot analysis, we compared the migration pattern of the PK-resistant PrP 27–30 bands of the L-type BSE prion strain used for this study and the two other BSE strains, C- and H-type BSE. Brain homogenates from C-, L-, and H-type BSE-infected Tg4092 mice were treated with 50 μg/ml PK and examined in a Western blot using the anti-PrP D15.15 monoclonal antibody. As expected, the three BSE isolates exhibited distinct banding patterns, resembling previous reports on C-, L-, and H-type BSE strains [7,9]. For example, as previously described [9], compared to classical BSE, atypical H- and L-type BSE showed higher and lower molecular mass bands for the unglycosylated prion protein, respectively (Fig 1E). Therefore, these data indicate that the strain features of the L-type BSE sample are maintained after passage in Tg4092 mice.

## Transmission electron microscopy of L-type BSE fibrils

Following the successful purification of the L-type BSE prions from the Tg4092 mouse brains, we sought to examine the morphology of the infectious protein samples using transmission electron microscopy (TEM). We performed negative stain electron microscopy tests on all fractions from the sucrose gradient purification as well as the P1 pellet sample of the PTA precipitation step, which was obtained before loading onto the sucrose gradient column (Fig 1). The P1 pellet was very crowded with amyloid fibrils as well as amorphous aggregates and lipid containing particles when observed by electron microscopy, indicating the need for additional

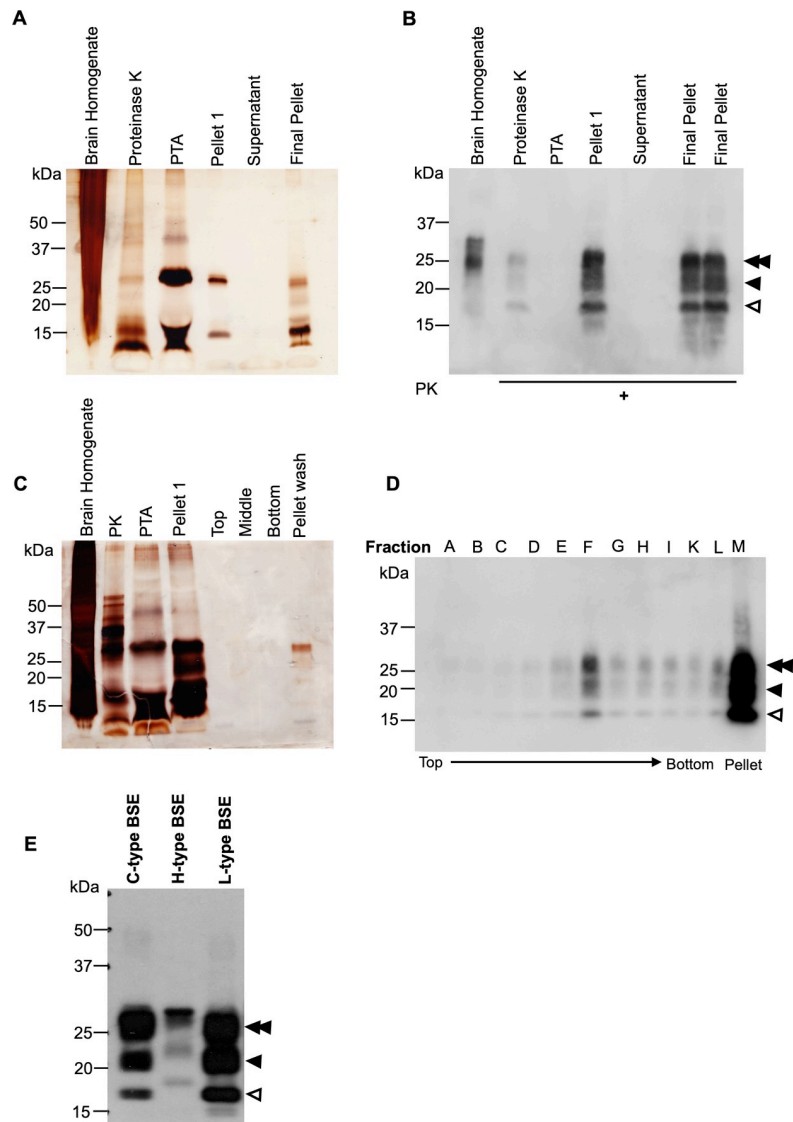

**Fig 1. Purification of L-type BSE prions.** (**A**) Silver stain SDS-PAGE gel of samples taken at different steps of the PTA purification. (**B**) Western blot of the L-type BSE prion purification samples using the D15.15 anti-prion monoclonal antibody. The PK-digested pellet 1 and the final pellet show the typical three bands of the bovine prion protein corresponding to di-, mono- and non-glycosylated forms (double arrowhead, single arrowhead, and open arrowhead, respectively). The two lanes labeled as final pellet represent the same sample. (**C**) Silver stain SDS-PAGE gel of samples from the sucrose step gradient centrifugation. The samples from the top, middle, bottom, and pellet wash fractions demonstrate the improved purity of the samples, even though the prion protein signal is relatively weak. Consistent with the Western blot analysis of the sucrose gradient fractions, the pellet-wash fraction shows stronger signals. (**D**) Western blot of the sucrose step gradient fractions. After ultracentrifugation through 40% and 80% sucrose, all fractions were separated based on their densities from top to bottom (A-L). The pellet-wash fraction (M) was obtained by washing the tube with 100 μl of sucrose buffer. The Western blotting was developed using the D15.15 antibody. As expected, the middle fraction (F) shows a stronger signal compared to the top (40%) and the bottom fractions (80%). The pellet-wash fraction (M) also showed a high yield of L-type BSE prions. The di-, mono-, and non-glycosylated bands are again indicated by a double arrowhead, single arrowhead, and open arrowhead, respectively. (**E**) Differences in migration patterns between L-type, C-type, and H-type BSE strains visualized in a Western blot of PK-treated brain homogenates of L-, C- and H-type BSE prions. Signals were detected using an anti-PrP monoclonal antibody, D15.15. In atypical BSE isolates, the electrophoretic migration of the unglycosylated PrP band is slightly higher (H-type) or lower (L-type) compared to classical BSE (C-type). The di-, mono-, and non-glycosylated bands are again indicated by a double arrowhead, single arrowhead, and open arrowhead, respectively.

purification steps (S1 Fig). Only a few amyloid fibrils were observed in the samples from the top and the bottom fractions, which yielded weak bands in the Western blot (Fig 1D), while the remaining fractions displayed no or only rare amyloid fibrils. Predominantly, fibrils were found in the negatively stained samples of the middle and the pellet-wash fractions. However, the micrographs of the pellet-wash fraction revealed a significantly higher yield of amyloid fibrils, in good correlation with the signal intensity in the Western blot. As expected, electron micrographs of samples from the sucrose gradient ultracentrifugation showed better purity compared to the pre-spin P1 sample and contained more clean, non-overlapping isolated fibrils and fewer amorphous aggregates or lipid particles. A more extensive examination of the samples revealed some non-fibrillar particles along with the fibrils (Fig 2), including small two-dimensional (2D) crystals [39].

## One- and two-protofilament L-type BSE fibrils

Overall, the L-type BSE fibrils were found to be morphologically heterogeneous when studied by EM. We found two main populations of amyloid fibrils: two-protofilament fibrils displaying two fibrillar densities with a gap between them (Fig 3A) and thinner single-protofilament fibrils (Fig 3B). Both fibrillar types exhibited helical properties. The one-protofilament fibrils, accounting for ~73% of the total fibril population, were significantly more abundant compared to the two-protofilament fibrils, which were identified in ~27% of the fibrils (Table 1). In contrast, in a previous study of a GPI-anchorless prion, the sample contained only helical fibrils containing two protofilaments [24]. While the two-protofilament morphology is in agreement with the conclusions drawn from previous studies on other prion strains [23,24,40,41], here, for the first time, we observed the single protofilament fibrils as the dominant structure of infectious L-type BSE prion fibrils. This quaternary structural arrangement had not been observed in previous electron microscopy investigations of infectious prion fibrils. However, in a study using atomic force microscopy (AFM) on ME7, 22L, and RML prions, most fibrils were composed of two protofilaments while some one-protofilament fibrils were also reported [40].

By analyzing hundreds of electron micrographs, we discovered a few interesting images in which the fibrils transitioned between one- and two-protofilament morphologies (Fig 4). Each of the six representative electron micrographs contains a long L-type BSE fibril that on one end consists of two protofilaments (white arrowheads), while the other end is split into two one-protofilament fibrils (black arrows). This finding appears to indicate the interaction of two separate one-protofilament fibrils to form a two-protofilament fibril. These micrographs preclude the possibility that the observed gap between the protofilaments is solely the result of a stain artefact, as the protofilaments appear to separate into distinct fibrils (Fig 4). In this context, a recent study demonstrated that the negative staining process or stain artifacts do not

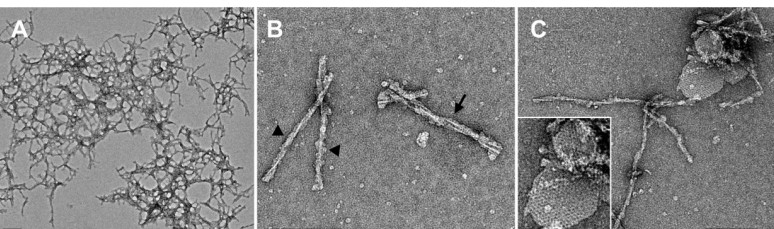

**Fig 2. Negative stain electron micrographs of purified L-type BSE samples showing heterogeneous morphologies.**
Representative electron micrographs of (**A**) large PrP 27–30 aggregates, (**B**) isolated forms of both single (arrowheads) and double (arrow) protofilament fibrils, and (**C**) 2D crystals along with amyloid fibrils. Grids stained with 2% uranyl acetate. Scale bar = 100 nm.

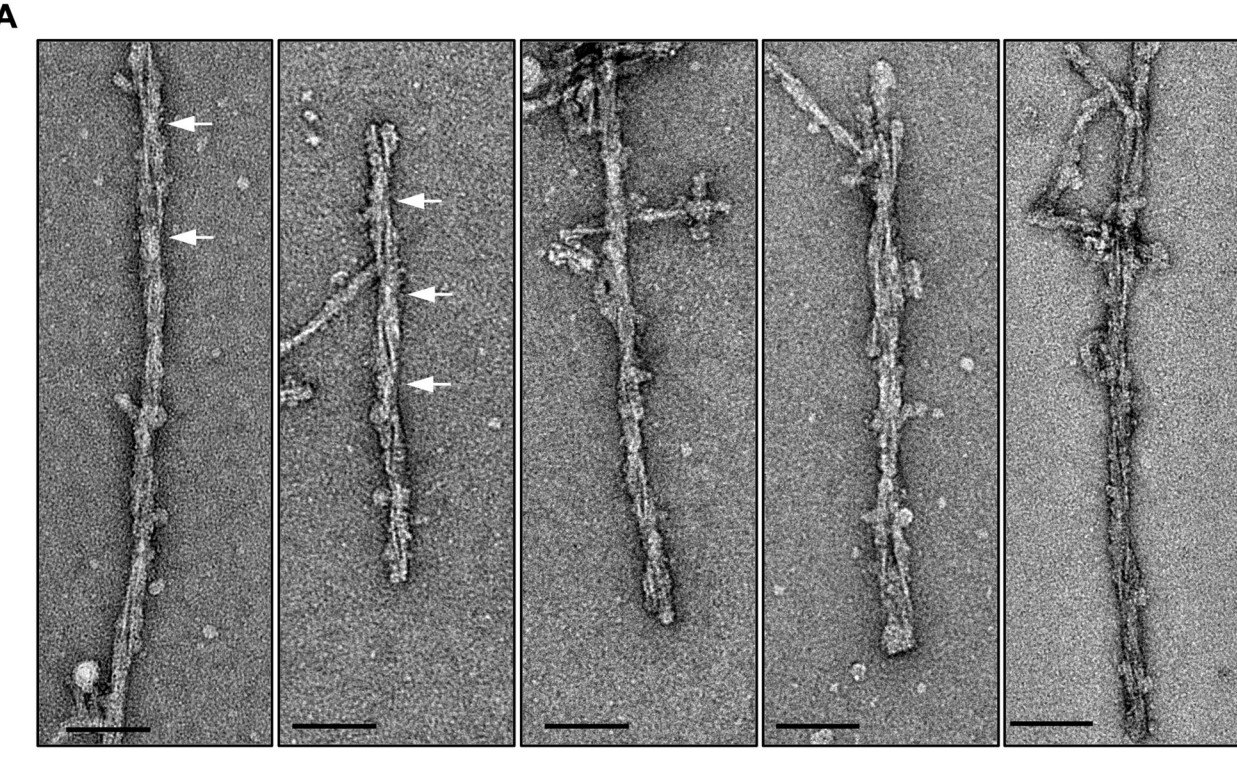

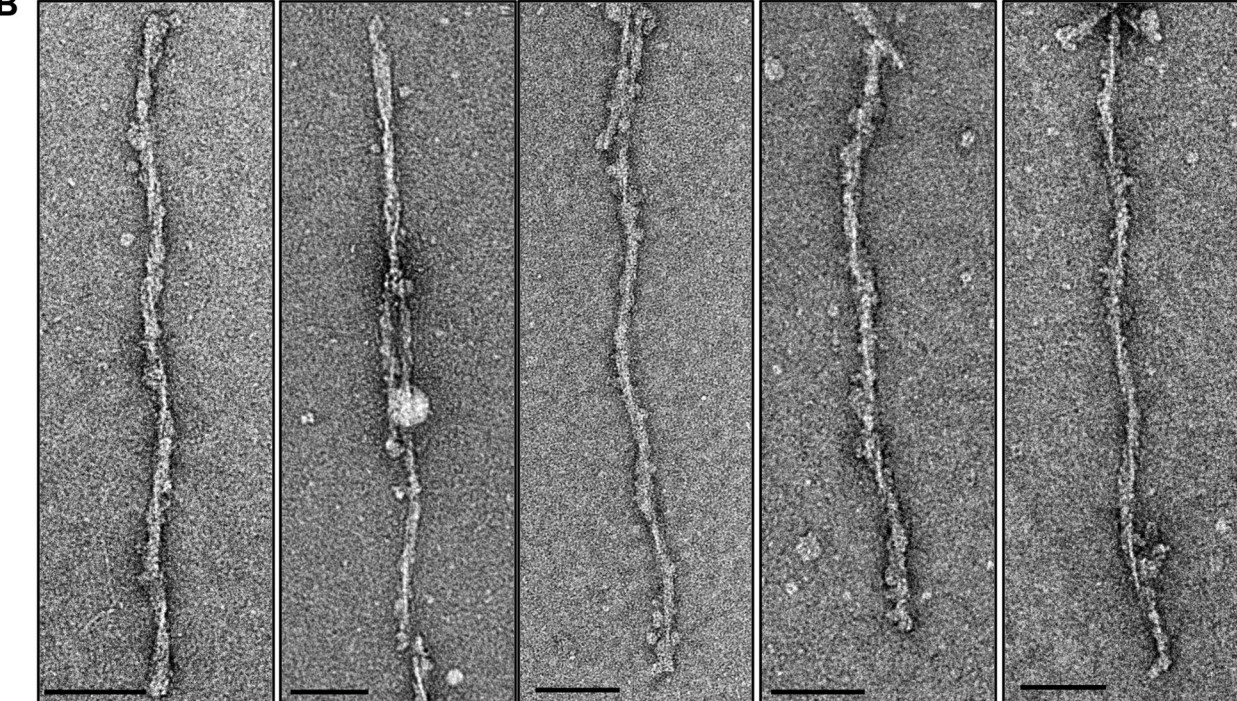

**Fig 3. Gallery of negatively stained L-type BSE fibrils.** Representative electron micrographs of L-type BSE fibrils with distinct morphologies containing two- (**A**) and one-protofilament (**B**) fibrils. The crossover regions for selected two-protofilament fibrils are shown using white arrows. Grids stained with 2% uranyl acetate. Scale bar = 100 nm.

**Table 1. Fibril dimensions of negatively stained brain-derived infectious L-BSE fibrils.**

| Morphology | Number of Fibrils | Width Mean ± SD (nm) | Helical Pitch Mean ± SD (nm) | Length Mean ± SD (nm) |
|---|---|---|---|---|
| Two-protofilament | n = 231 fibrils | 22.5 ± 3.4 | 102.3 ± 16.9 | 598.2 ± 268.5 |
| One-protofilament | n = 636 fibrils | 10.6 ± 1.7 | 89.3 ± 15 | 387.5 ± 155.4 |

The measurements were performed using EMAN' boxer image software.

substantially alter the dimensions and overall conformation of amyloid fibrils based on the observation that the morphology of mouse-adapted prion fibrils in a negative stain experiment were in agreement with cryo-EM and atomic force micrographs of the same samples [30,42].

As shown in Fig 5, from a selection of 867 fibrils, the average diameter of the two-protofilament fibrils at their widest part was 22.5 ± 3.4 nm, which resembles the width of the double-helical fibrils of infectious RML prion fibrils published in a recent study using cryo-EM and AFM microscopies [30]. The average width of one-protofilament L-type BSE fibrils was

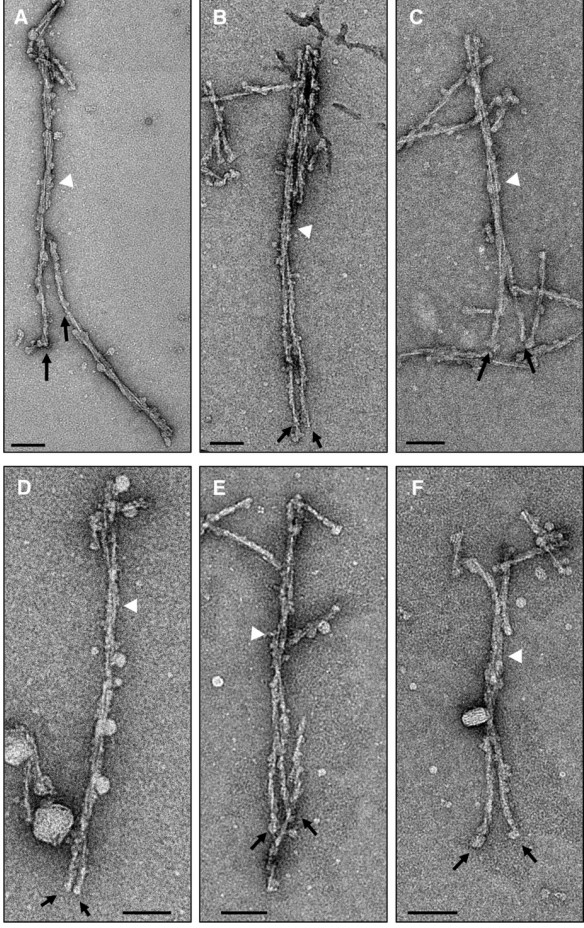

**Fig 4. Side-by-side comparisons of the one- and two-protofilament fibrils. (A-F)** Negative stain electron micrographs showing the two distinct morphologies of L-type BSE fibrils side-by-side. (**A**) A negative-stained electron micrograph showing a long, isolated fibril composed of two protofilaments (white arrowhead), which appears to be broken and split into two one-protofilament fibrils towards its lower end (black arrows). (**B-F**) L-type BSE amyloid fibrils showing a two-protofilament fibril at one end (white arrowheads) and two one-protofilament fibrils at the other end (black arrows). Scale bar = 100 nm.

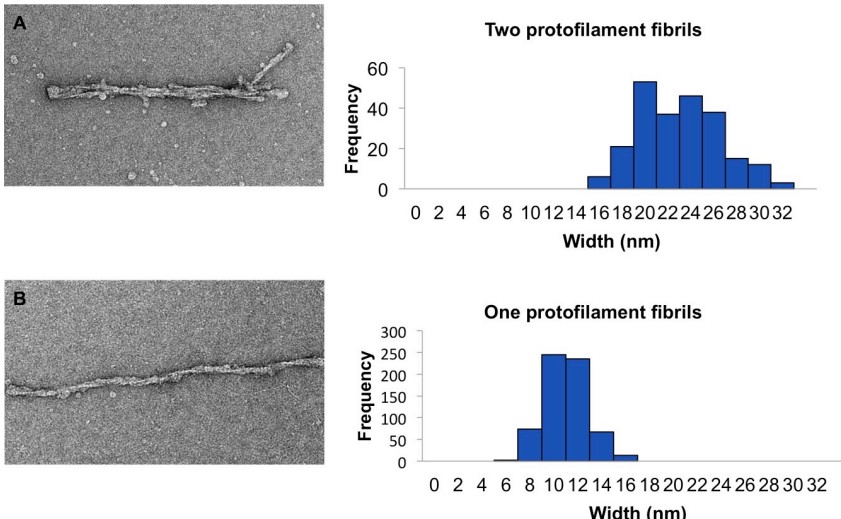

**Fig 5. Width distribution of L-type BSE fibrils.** (**A**) A sample electron micrograph and histogram of the maximum width of 231 two-protofilament L-type BSE fibrils. (**B**) A sample electron micrograph and histogram of the maximum width of 636 one-protofilament L-type BSE fibrils. The width measurements were performed using EMAN's boxer program.

10.6 ± 1.7 nm at the widest region of the fibrils. It is notable from the histogram (Fig 5) that the width of the two-protofilament fibrils is about twice that of one-protofilament fibrils, plus the material or gap between the two protofilaments can account for the slightly larger values for these thicker fibrils. In a previous cryo-EM study, the two-protofilament fibrils of GPI-anchorless mouse PrP$^{Sc}$ had a maximum width of ~10 nm [24]. The underglycosylated and GPI-anchorless PK-resistant PrP$^{Sc}$ fragment was found to migrate as a single band with a mass of ~17 kDa [43], while the PK-resistant fragments of L-type BSE PrP 27–30 migrated between 30 and 17 kDa, depending on their glycosylation (Fig 1B and 1D). The mass difference between these two prion variants can easily explain the apparent difference in fibril widths. It should also be noted that in the recent PIRIBS-based 263K prion structure the widths of the fibrils were reported as 13–20 nm [28], which is roughly comparable with the widths of our one- and two-protofilament fibrils.

The helical pitch for one- and two-protofilament fibrils were on average 89 nm and 102 nm, respectively (Table 1). These values were both in the range that was reported for the distance per half turn for hamster-adapted scrapie associated fibrils (SAFs) [40]. We also measured the length of the fibrils to find out whether there is a difference in both morphologies. The negatively stained two-protofilament fibrils had an average length of 598 nm, and the single-filament fibrils were on average 387 nm in length. The greater length for the first group may to be due to a reduced sensitivity of these thicker (wider) fibrils to breakages (Table 1).

### Image processing and 3D reconstructions

In order to decipher the ultrastructure of L-type BSE prions, we applied image processing techniques on representative negative-stain electron micrographs. The quality and good yield of isolated, non-overlapping L-type BSE fibrils enabled us to employ image processing and 3D reconstruction strategies on individual amyloid fibrils, which provided insightful results.

The first approach was to generate 2D class averages from hundreds of short image sections from isolated fibrils, which were treated as "single particles" during the image processing. The class averages on a group of 2D images contain higher signal-to-noise ratio compared to a

single raw image, further intensifying the features in the object of interest, which aid the interpretation of the images [44]. Reference-free class averaging was performed independently on each group of one- and two-protofilament fibrils. In a segmentation procedure, boxes of 200 by 200 pixels (corresponding to 61.4 by 61.4 nm) with a 50% overlap were applied along the fibril axis, and the resulting segmented particles were separately averaged and sorted into different classes according to structural similarity. As depicted in Fig 6A–6C, class averages obtained from the thicker, two-protofilament fibrils revealed two fibrillar densities with an apparent, stain-filled space between them. Given that the particles were sorted into classes based on structural similarities, class A appears to be related to the crossover section of the helical fibrils. The same approach on the thinner fibrils revealed class averages exhibiting a helical fibril with one protofilament only (Fig 6D–6F), that are also characterized by the narrower fibril width and the lack of a separating gap as seen in Fig 6A–6C.

The helical nature of L-type BSE fibrils allowed us to perform three-dimensional (3D) helical reconstructions on individual fibrils. We performed this technique on fibrils with an apparent helical twist covering at least one half helical turn (180°), following an earlier protocol [24]. The 3D maps showed two distinct morphologies of the reconstructed fibrils, including fibrils with two intertwined protofilaments twisting around the fibril axis (Fig 7A) and single protofilament fibrils (Fig 7B). In the process of extended helical reconstructions, reconstructed fibrils were rotated and averaged along the fibril axis to generate an average cross-section of the fibril. The averaged cross-section densities of the first group contain two apparent densities

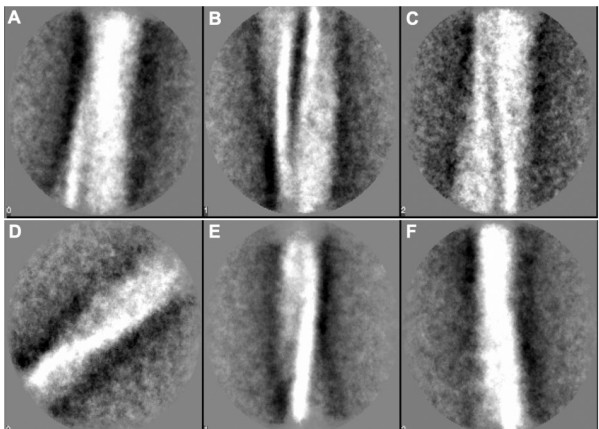

**Two-protofilament
L-type BSE fibrils**

**One-protofilament
L-type BSE fibrils**

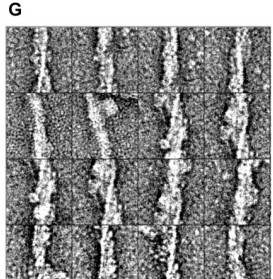 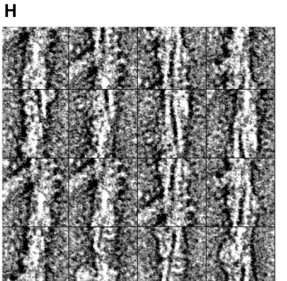

**Fig 6. Alignment, classification, and averaging of L-type BSE fibrils.** (**A-C**) Reference-free alignment and classification of particles of two-protofilament L-type BSE fibrils. Class averages of 297 segmented particles exhibiting different regions of the fibrils, including the crossover region (**A**), showing two intertwined protofilaments with an apparent, stain-filled gap between them. (**D-F**) Class averages from the reference-free alignment of 532 particles of one-protofilament L-type BSE prion fibrils. All three class averages show a single helical filament only. Representative images of aligned fibril segments of one- (**G**) and (**H**) two-protofilament fibrils before classification.

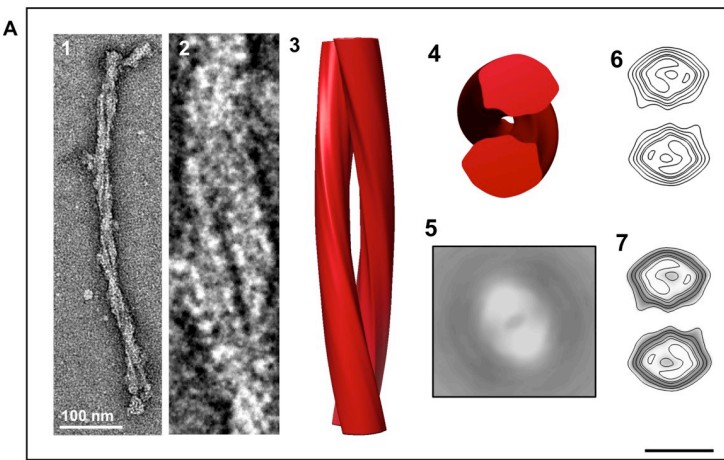

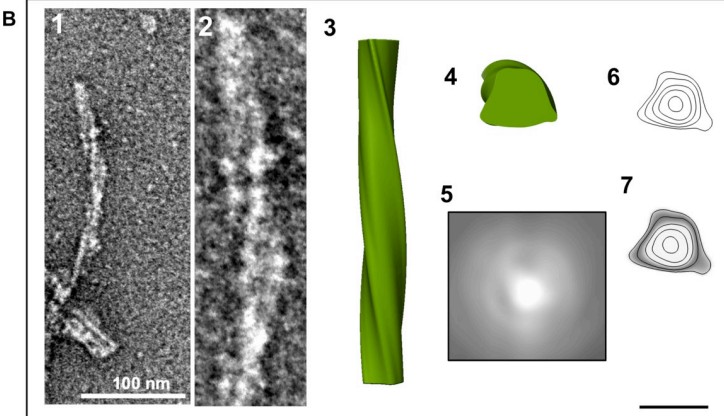

**Fig 7. Three-dimensional reconstructions of L-type BSE prion fibrils.** (**A**) A 3D reconstruction of a two-protofilament fibril. (**1**) A negatively stained electron micrograph of a L-type BSE fibril, and (**2**) magnified view of the selected fibril section used for generating the 3D reconstruction. (**3**) A surface view of the 3D reconstruction showing two intertwined helical protofilaments. (**4**) Cross-section of the reconstruction. (**5**) The corresponding 2D projection of the averaged view of the cross-section region. (**6**) A contour map of the cross-section. (**7**) A contour density plot of the cross-section obtained by superimposition of the contour map onto a cross-section from the 3D volume. (**B**) A 3D reconstruction of a one-protofilament fibril. (**1**) A negatively stained electron micrograph of a L-type BSE fibril, and (**2**) enlarged view of the selected fibril. (**3**) Surface view of the 3D reconstruction. (**4**) Cross-section of the reconstruction. (**5**) The corresponding 2D projection of the averaged view of the cross-section region. (**6**) A contour map of the cross-section. (**7**) A contour density plot of the cross-section. The 3D reconstructions were low-pass filtered to 20 Å resolution. Scale bar = 10 nm.

corresponding to two protofilaments (Fig 7A5), whereas the 2D projection of the averaged cross-section of the second group includes only one fibrillar density, i.e. one protofilament (Fig 7B5). This observation confirms the existence of both distinct morphologies, which had been inferred from the fibril diameters (Fig 5 and Table 1) and individual micrographs (Figs 3 and 4). Contour maps and contour density maps of both reconstructions (Fig 7A6, 7A7, 7B6 and 7B7) provided us with more information about the ultrastructural arrangements of these amyloid fibrils. The density distribution revealed distinct densities in both morphologies, which could correspond to the different components of the protein (peptide, glycans, GPI-anchor).

We collected about 700 electron micrographs of L-type BSE fibrils, which enabled us to produce a panel of 3D helical reconstructions on isolated two- (Fig 8A and 8B) and one-

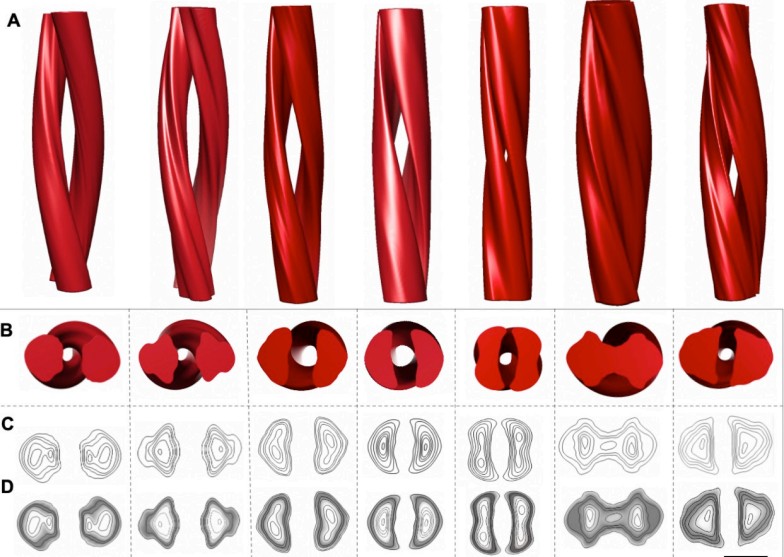

**Fig 8. Gallery of independent, 3D reconstructions of two-protofilament L-type BSE fibrils.** (**A**) 3D volumes of different two-protofilament L-type BSE fibrils. (**B**) Cross-section views of the helical reconstructions. (**C**) Contour maps and (**D**) density maps of the reconstructed volumes. Scale bar = 10 nm.

protofilament fibrils (Fig 9A and 9B). Comparing the panel of separate 3D reconstructions provided further evidence regarding the heterogeneity and quaternary structural arrangements of these amyloid fibrils. The cross-sections of all 3D reconstructed fibrils, although of low resolution, adopted a triangular or roughly oval morphology, similar to the shape of cross-sections of other β-solenoid proteins [45,46]. Representative 3D reconstructions of one- and two-protofilament L-type BSE fibrils were superimposed with the recent PIRIBS structure [28] and 4RβS model, with N-linked glycans attached [24,25] (Fig 10). However, it should be noted that

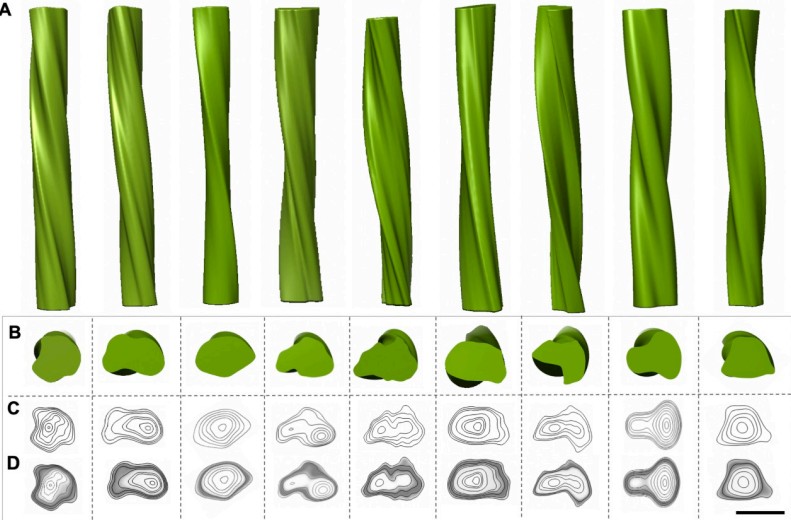

**Fig 9. Gallery of independent, 3D reconstructions of one-protofilament L-type BSE fibrils.** (**A**) 3D volumes of various thin, one-protofilament L-type BSE fibrils. (**B**) Cross-section views of the helical reconstructions. (**C**) Contour maps and (**D**) density maps of the reconstructed volumes. Scale bar = 10 nm.

the placeholders for the glycans underestimate the size and heterogeneous nature of the complex carbohydrates. While the attachment point for the GPI-anchor was observed in the PIR-IBS structure [28], the GPI-anchor is missing in both overlays (Fig 10). Another factor that influences the size comparison is the lack of the hydration shell that surrounds all proteins. Both the PIRIBS structure and the 4RβS model are shown 'naked', which again underestimates their diameters. As a consequence, the 4RβS model appears to be a substantially better fit for the observed L-type BSE fibril cross-sections than the elongated PIRIBS cryo-EM structure that was reported using 263K hamster prions (Fig 10). However, a more compact PIRIBS structure might fit into the envelopes of both the one- and two-protofilament L-type BSE fibrils.

## Immunogold labeling for N- and C-terminal epitopes

To determine whether the fibrils contain PrP 27–30 or more truncated forms of the prion protein, we performed immunogold labeling experiments using different anti-PrP Fab fragments and monoclonal antibodies targeting epitopes located in different regions of the prion protein,

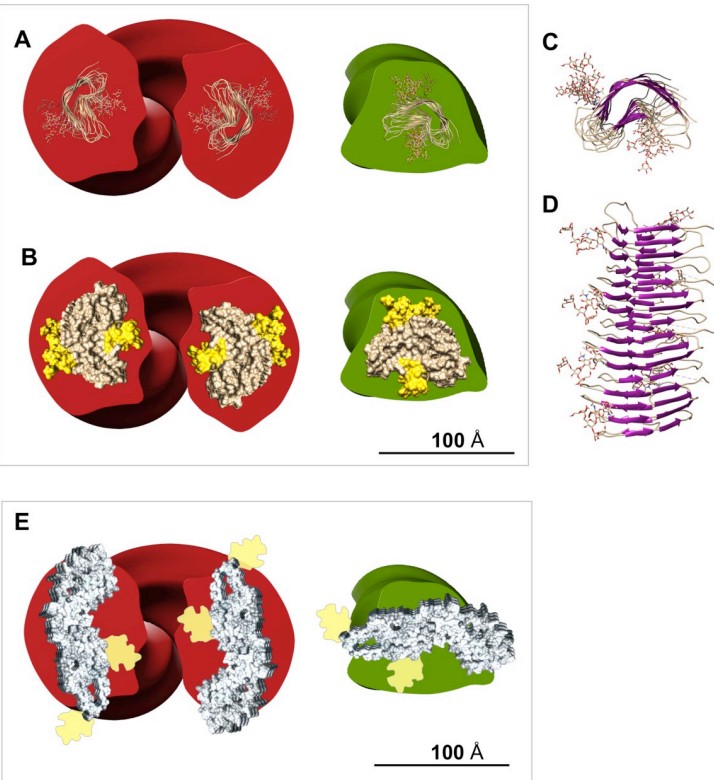

**Fig 10. Superimposition of the 3D L-type BSE fibril reconstructions with the proposed structures for PrP<sup>Sc</sup>.** (**A**) A representative view of the 4RβS model of PrP<sup>Sc</sup> with attached N-linked glycans superimposed onto the cross-section view of 3D reconstructions of a two-protofilament (red) and a one-protofilament (green) L-type BSE fibril. The size of the 4RβS model (50 Å x 30 Å) was set based on the previous cryo-EM and molecular dynamics simulation studies [24,25]. (**B**) A surface view of the 4RβS PrP<sup>Sc</sup> model carrying glycans superimposed onto the cross-section view of 3D reconstructions of a two-protofilament (red) and a one-protofilament (green) L-type BSE fibril. (**C**) Top and (**D**) side views of a representative tetrameric version of the 4RβS PrP<sup>Sc</sup> model containing glycans [25]. (**E**) Superimposition of the cross-section of a two-protofilament (red) and a one-protofilament (green) L-type BSE fibril and the PIRIBS structure obtained from the recent cryo-EM study on brain-derived 263K prions [28]. The size of the PIRIBS structure (13 nm) was set as indicated in the pre-print. The space-filling view shows the protein moiety only. Estimated densities representing N-linked glycans are attached to the structure in positions indicated in the pre-print [28].

including N-terminal and C-terminal epitopes. For this purpose, we first employed two recombinantly produced antibody fragments: Fab 69, which detects an epitope within residues $_{100}$GWGQGGTHGQW$_{110}$ near the N-terminus of truncated bovine PrP 27–30, and Fab 29, which binds an epitope within residues $_{227}$TQYQRESQAYY$_{237}$ near the very C-terminus of bovine PrP (based on the numbering of six octapeptide-repeat PrP). The best immunogold labeling was achieved when the purified samples were pre-treated by a denaturing procedure with urea, suggesting a reduced accessibility of these epitopes in the native state. Both one- and two-protofilament fibrils were decorated by the Fab fragments, followed with a secondary anti-Fab antibody and a tertiary antibody carrying 5 nm gold particles (Fig 11). Control experiments were run concurrently without use of the primary Fab fragments, which showed no or very rare presence of gold particles, demonstrating the specificity of the immunogold labeling (Fig 11D and 11H).

In another attempt at immunogold labeling we used YEG mAb Sc-G1, a novel monoclonal antibody that recognizes a conformational epitope on native PrP$^{Sc}$ only (Fang et al., manuscript in preparation). By using this monoclonal antibody conjugated with 6 nm gold particles, we were able to label L-type BSE prions in their native form (Fig 12A and 12B). In contrast, no antibody decoration occurred in the control samples (Fig 12C), indicating the specificity of the labeling. Successful labeling of the fibrils in the purified L-type BSE samples allowed us to

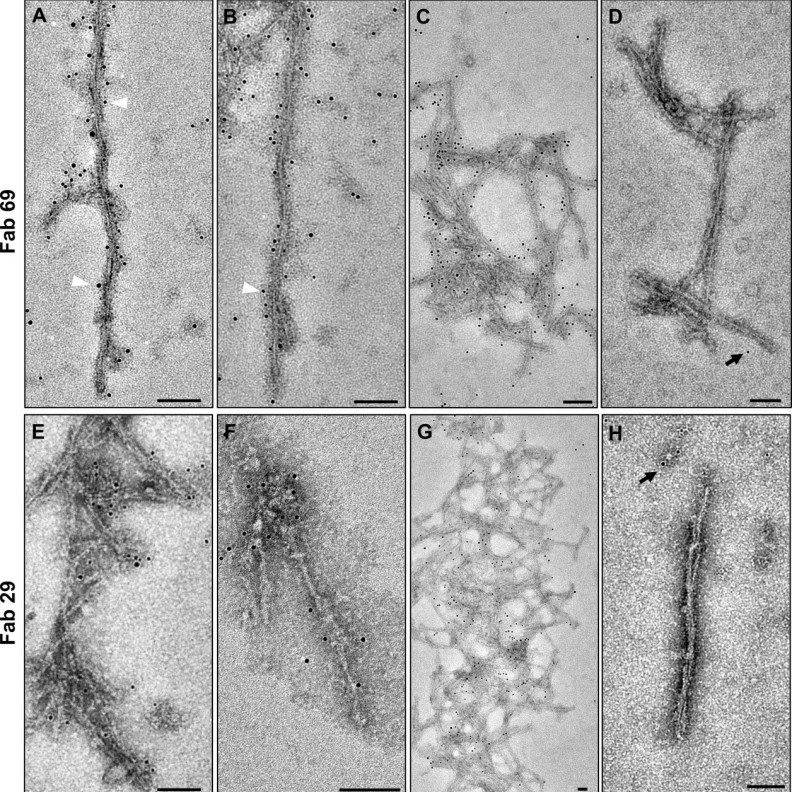

**Fig 11. Immunogold electron microscopy of purified L-type BSE fibrils.** Decoration of different fibrillar assemblies, including (**A**) two-protofilament, (**B**) one-protofilament, and (**C**) fibrillar aggregates, with Fab 69 and a 5 nm gold-conjugated ternary detection system. White arrowheads highlight a few gold particles along the fibrils. (**D**) Grids that were incubated without primary antibody showed only rare gold particles (black arrow), demonstrating the specificity of the Fab 69 labeling. Immunogold labeling of (**E**) two-protofilament, (**F**) one-protofilament, and (**G**) aggregates of L-type BSE fibrils with Fab 29 and a 5 nm gold-conjugated ternary detection system. (**H**) No specific labeling was observed in the grids with no primary antibody. Scale bar = 100 nm.

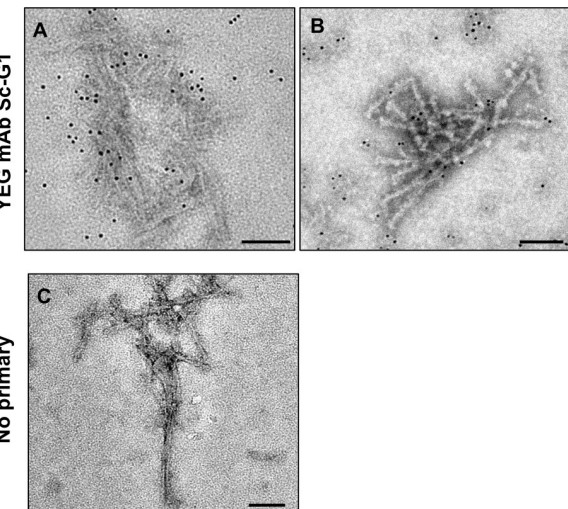

**Fig 12. Immunogold labeling of purified L-type BSE fibrils with a conformation-dependent monoclonal antibody.** (**A & B**) Represents samples labeled with the YEG mAb Sc-G1 monoclonal antibody, which recognizes a discontinuous epitope on native PrPSc only. The monoclonal antibody was detected with a secondary antibody coupled to 6 nm gold particles (**C**) Control grid that underwent the same treatment except for the omission of the primary antibody, showing no gold labeling. Scale bar = 100 nm.

conclude that our isolated L-type BSE fibrils contain native PrP 27–30. In addition to the two Fab fragments that detect epitopes at both N- and C-terminal regions, respectively, the positive immunogold labeling result with YEG mAb Sc-G1 confirms that the observed L-type BSE fibrils are generated by polymerization of PrP 27–30 monomers.

## Infectivity of purified BSE prions

To confirm whether the purified L-type BSE samples are infectious, we inoculated Tg4092 mice that overexpress bovine PrPC [31] with samples from the pellet 1 and the PK-treated final pellet. One group of transgenic mice was also inoculated with 1% brain-homogenate without PK treatment. L-type BSE-infected Tg4092 mice were euthanized upon developing clinical symptoms. In all three groups, we observed clinical symptoms of prion disease, confirming that these preparations contain high levels of infectivity. Moreover, as shown in Table 2, the mice inoculated with the final pellet showed the shortest incubation period of about 206 days compared to those inoculated with the semi-purified P1 pellet and the brain homogenate, which caused disease in 240 days and 227 days, respectively. Therefore, we were able to conclude that the isolated samples contain infectious L-type BSE prions and that the purification protocol and digestion with PK did not destroy the infectivity. By using a standard curve that relates the incubation period to the prion titer, we were able to calculate the $LogID_{50}$/mL units for each sample [47] (Table 2). However, it should be taken into account that the calibration was originally performed using homozygous Tg4092 animals infected with C-type BSE prions, while we worked with

**Table 2. Infectivity of the purified L-type BSE samples.** Intracerebral inoculation of transgenic mice with the BSE prion samples obtained during purification experiments.

| Mouse Line | Sick/Total | BSE Strain | Inoculum | Incubation Time (Days ± SEM) | Prion Titers ($LogID_{50}$/ml) |
|---|---|---|---|---|---|
| Tg4092 | 11/11 | L-type | Brain homogenate | 227 ± 15 | 7.31 |
| Tg4092 | 5/5 | L-type | Pellet 1 | 240 ± 1.6 | 6.92 |
| Tg4092 | 12/12 | L-type | Final pellet | 206 ± 3.4 | 8.06 |

hemizygous Tg4092 animals assaying L-type BSE prions. Therefore the titers in Table 2 should be considered a lower estimate for the true titer of the L-type BSE samples.

## Discussion

In this study, we employed electron microscopy and image processing techniques to analyze infectious, brain-derived L-type BSE amyloid fibrils, which included the typical posttranslational modifications: N-linked glycans and a GPI-anchor. We discovered that the L-type BSE prions present as long amyloid fibrils, which matches with previous pathological studies that indicated the increased amyloidogenicity of this strain when compared to the classical (C-type) BSE strain [8]. The purified samples presented different conformational states of the infectious prion protein, such as small amorphous aggregates, 2D crystals, isolated amyloid fibrils, and clumps of fibrillar aggregates. Intracerebral inoculation of the purified samples into Tg4092 mice produced clinical prion disease symptoms, confirming that the purified bovine PrP 27–30 maintained its infectious property and is capable of converting normal PrP into its abnormal disease-causing conformer.

Our purification protocol allowed the isolation of a high yield of L-type BSE prion fibrils, which proved to be morphologically heterogeneous when examined by transmission electron microscopy. The observed heterogeneity agrees with a previous cryo-EM study of anchorless prion fibrils [24]. In this study, we discovered two distinct morphological populations of L-type BSE amyloid fibrils. The majority of fibrils were observed to be single-protofilament fibrils, with an average width of 10.6 nm. These single-protofilament fibrils accounted for ~73% of the total fibril population. The second group included thicker two-protofilament fibrils with an average diameter of 22.5 nm, which is about twice the width of the single-proto-filament group.

The two-protofilament morphology was consistent with the previous cryo-EM and image processing study on GPI-anchorless RML prion rods, in which each PrP 27–30 fibril was comprised of a twisting pair of intertwined protofilaments [24]. Previous negative stain EM and cryo-EM studies of other prion strains, such as ME7, 22L, and RML, also reported the presence of two-protofilament fibrils [23,30,40,42]. However, we discovered that mature L-type BSE fibrils are composed of one-protofilament and two-protofilament fibrils in the same samples. We verified this finding by 2D class average and 3D reconstruction techniques. Furthermore, immunogold labeling experiments using Fab fragments that detect epitopes within residues $_{100}$GWGQGGTHGQW$_{110}$ and $_{227}$TQYQRESQAYY$_{237}$ demonstrated that both ends of bovine PrP 27–30 are present in the one- and two-protofilament fibrils.

The cross-sections of the L-type BSE fibril reconstructions (Figs 8B–8D and 9B–9D) strongly resemble the ones observed for the GPI-anchorless PrP$^{Sc}$ fibrils proposed to have a 4RβS structure [24]. Moreover, our image processing results showed that the average cross-section of each protofilament from both the one- and two-protofilament fibrils adopts a triangular or elliptical shape, which is akin to the cross-section of amyloids from β-solenoidal proteins [45,46]. In the 4RβS arrangement, conformational differences between prion strains are believed to be related to minor variations in the threading of the PrP$^{Sc}$ monomers, more localized in the connecting loops and less affecting the β-strands [48]. A recent MD simulation study suggested that the 4RβS arrangement is flexible enough to allow such variations, explaining the occurrence of different prion strains [24–26,49]. Thus, the variations in the quaternary structure of bovine PrP$^{Sc}$, as compared to previously studied prion strains, could provide insights into the prion strain phenomenon [48].

However, recent evidence, including the recent cryo-EM study, proposes a PIRIBS arrangement for the structure of the infectious prion [28,29]. In the cryo-EM study on brain-derived

263K prions, the isolated fibrils were composed of one protofilament only, and the protein moiety alone had a width of 13 nm. The widths of the one-protofilament L-type BSE fibrils observed in our investigation are too narrow to match the width of the PIRIBS fibrils. In theory, the two-protofilament L-type BSE fibrils could encompass the width of the PIRIBS structure, however, the observation of individual fibrils with a two-protofilament structure on one end and two individual one-protofilament fibrils at the other appears to be inconsistent with such an interpretation (Fig 4). Nevertheless, the heterogeneity in our samples and in previous investigations [24] suggests that more than one structure could co-exist in the brain of one host. Moreover, in a previous study by Spagnolli et al., it was shown that, through deformed templating, one end of fibrils with PIRIBS structure could template a 4RβS PrP$^{Sc}$ and vice versa [50]. Further high-resolution studies of BSE prions are needed before a definitive conclusion can be made on this point.

In conclusion, our structural analysis of infectious, brain-derived L-type BSE fibrils provided novel insights about the biochemical and conformational characteristics of atypical BSE prions. The presented data offered insights into the quaternary structure of BSE amyloid fibrils, provided a framework for new hypotheses on how prion proteins interact, assemble into amyloid fibrils, and encrypt prion strains. Investigating strain-specific features of BSE prions could help to resolve the structure of these pathogens and accelerate the development of prion vaccines and other countermeasures.

## Methods

### Ethics statement

The BSE transmission experiments were carried out at the Canadian Food Inspection Agency, Lethbridge Laboratory, Lethbridge, Alberta, in accordance with guidelines set by the Canadian Council on Animal Care and approved by the animal care use committee for the CFIA-ADRI Lethbridge Laboratory (ACC#0902). All bioassay experiments in Tg4092 mice were carried out at the Centre for Prions and Protein Folding Diseases (CPPFD), University of Alberta, in accordance with guidelines set by the Canadian Council on Animal Care and approved by the animal care use committee for Health Sciences 2 (protocol AUP00000884).

### L-type BSE prions

Brain samples from the single L-type BSE affected cow discovered in Canada were confirmed to be BSE positive at the Canadian National BSE Reference Laboratory. The brain tissue from this animal was passaged into two calves. For this purpose the colliculus tissue was homogenized to 10% (w/v), sonicated, and spun to remove large cellular debris. The supernatant was aspirated into a syringe for subsequent injection into the cranial cavity of two steer calves at approximately 5 months of age (1 mL of 10% homogenate per steer). Prior to inclusion in this project, steers of interest were genotyped to ensure no abnormalities were present in the prion gene. Once confirmed as genetically normal, the steers were moved into the biosafety level 3 containment pens at the CFIA Lethbridge Laboratory. Following challenge, the animals were monitored regularly for the appearance of clinical signs. At 16 months post challenge, both steers began to show mild clinical signs of BSE infection. With noted progression over the next several weeks, the steers were euthanized at 17 and 18 months post challenge, respectively. The brain stem was tested immediately using the Prionics rapid test platform and generated OD values of >8,000 for both animals. The kit lot positive/negative cutoff OD value was 113 with >95% of BSE negative samples giving an OD value of 0 using this kit lot. Both of these sample homogenates were also strongly positive on a Western blot with three distinct immuno-reactive glycoforms ranging in size from 28 kDa down to 18 kDa.

To type the protease-resistant prions that were found in the central nervous system tissue from the two challenged steers, the homogenate was run on a hybrid Western blot. This assay utilizes mild and stringent protease digestion, a panel of antibodies with different epitopes and molecular weight/glycoform ratios to type BSE samples. Classical BSE PrP$^{Sc}$ has a ~2:1 ratio of di- to mono-glycosylated isoforms when detected on western blot, but the two L-type BSE challenged steer samples had ratios much closer to 1:1; this is commonly seen in atypical BSE. Proteinase K digestion under stringent conditions caused a significant decrease in PrP$^{Sc}$ detected, another consistent characteristic of atypical BSE prions. SDS-PAGE gel separation of the protease resistant core of PrP revealed a subtle 0.5 to 1 kDa molecular weight shift down when compared to the classical BSE control samples. Finally, the PrP$^{Sc}$ from the challenged steers was reactive with core anti-prion antibody 6H4 (Prionics AG, Switzerland) but was not reactive with the slightly more N terminal P4 anti-prion antibody (R-Biopharm AG, Germany). The molecular weight shift and antibody reactivity profile confirmed that the N-terminally truncated PrP$^{Sc}$ detected is L-type atypical BSE. These BSE typing results for the two challenged steers were consistent with the hybrid Western blot results for the Canadian L-type BSE field case used for the challenge.

## Transgenic mice and genotyping

Tg4092 mice [31] that overexpress the bovine prion protein were kindly provided by Dr. Stanley B. Prusiner (University of California, San Francisco). The presence of the transgene was identified by PCR using tail-derived genomic DNA, which was amplified using primers, forward 5'.TCGATCCAGAGCCTTTGAATTGAG.3'; and reverse 5'. GGGTGAAATGGTCAG TGCATTACG.3'. PCR thermocycling conditions were as follows: 94°C for 3 min, (94°C for 20 sec, 55°C for 30 sec, 72°C for 90 sec) repeated 35 times, 72°C for 1 min, hold at 4°C, resulting in a 836 bp fragment that was visualized on a 1% agarose gel containing ethidium bromide.

## Bioassays

Hemizygous Tg4092 mice overexpressing the bovine prion protein [31] were inoculated intra-cerebrally with 30 μL of diluted L-type BSE bovine brain homogenate per animal for the primary production of L-type BSE prions. Subsequent bioassay experiments were performed in the same manner, again using Tg4092 mice. The first group of transgenic mice was inoculated intra-cerebrally with 30 μl of 1% brain homogenate samples from L-type BSE-infected Tg4092 mice. Two other groups were injected separately with 30 μl of 1% pellet 1, and the final pellet samples (PK-digested) obtained from the purification experiments. Following inoculation, animals were monitored regularly for the appearance of prion disease clinical symptoms such as weight loss, scruffy coat, loss of appetite, ataxia, and dyspnea. Once the terminal stage of the disease was reached, the animals were euthanized by $CO_2$ asphyxiation followed by cervical dislocation. The brains were collected and stored at -80°C for future analysis. Prion titers were calculated by using a standard curve that relates the incubation period in Tg4092 mice to the prion titer [47]. However, the calibration was performed using homozygous Tg4092 animals infected with C-type BSE prions. Therefore the bioassay results for our L-type BSE samples in hemizygous Tg4092 mice underestimates the true prion titer.

## Isolation of BSE prions

Infectious L-type BSE prions were isolated from the brains of terminally sick L-type BSE-infected Tg4092 mice using a previously developed protocol [36]. First, all brains were pooled and homogenized at 20% concentration (w/v) in phosphate-buffered saline (PBS). Four to ten brains were used in separate purification experiments. The 20% brain homogenate was then

clarified at 500 × g for 5 minutes, and the supernatant was collected and added to a new tube with an equal volume of 4% Sarkosyl in PBS to make 10% w/v brain homogenate. Then, the sample was aliquoted into 1 mL screw-cab microcentrifuge tubes and subjected to digestion with PK (50 μg/ml) at 37˚C for 1 hour. The PK treatment reaction was stopped with the addition of 10 mM Phenylmethylsulfonyl Fluoride (PMSF). The process continued by adding 2% sodium phosphotungstic acid (PTA, pH 7.2) (SIGMA P-6395) to the aliquots and overnight (16 hours) incubation at 37˚C. Afterwards, the samples underwent centrifugation at 16,000 × g for 30 minutes and the P1 pellet fraction was obtained and resuspended with 0.2% Sarkosyl in PBS. Subsequently, 2% Sarkosyl in PBS and 2% PTA were added to the resuspended pellets, and the samples were centrifuged again at 16,000 × g for 30 minutes to obtain the final pellet. The final pellet was resuspended with 0.2% Sarkosyl in PBS. Samples collected from different steps of the purification were stored at -80˚C for future analyses.

In order to generate highly purified L-type BSE prion isolates suitable for structural examination, we combined the PTA-precipitation protocol with a sucrose step-gradient centrifugation. In this method, the standard PTA purification was performed until the first pellet (P1) was obtained, and after resuspension with 0.2% Sarkosyl in PBS, the pellet 1 was loaded onto a sucrose-step gradient of 40% and 80% sucrose and subjected to ultracentrifugation at 115,000 × g at 4˚C for 16 hours (overnight). Following centrifugation, 500 μL fractions were collected from the top of the ultracentrifugation tube. The bottom of the tube was washed with 100 μl of sucrose buffer (10 mM Tris HCl pH 7, 1mM NaN$_3$, 0.2% Sarkosyl) to recover pelleted proteins and labeled as 'pellet wash'. All collected samples were stored at -80˚C for SDS-Page, silver staining and TEM studies.

## Western blotting

Samples were mixed with a gel-loading buffer (Bio-Rad) containing 2% (w/vol) SDS and heated at 100˚C for 10 minutes before electrophoresis. The proteins were separated by SDS-PAGE gel electrophoresis using 12% acrylamide gels (Bio-Rad), run for 1 hour at 150 volts and blotted electrically onto a polyvinylidene difluoride (PVDF) membrane (Millipore) at 100 volts for 1 hour. The blotted membranes were blocked with 5% (w/v) BSA in Tris buffered saline solution containing 0.05% Tween 20 (v/v) (TBST), overnight, at 4˚C and incubated with anti-PrP antibody, D15.15 at a 1:5,000 dilution, for 1 hour, at room temperature, followed by washing three times for 5 minutes with TBST. Next, the membranes were incubated with alkaline phosphatase conjugated anti-mouse IgG (Bio-Rad) at a 1:10,000 dilution in TBST for one hour and washed three times for 5 minutes in TBST. Prion protein signals were developed by adding ~1 ml alkaline phosphatase (AP) substrate (Bio-Rad) and detected by chemiluminescent visualization using ImageQuant (GE Life Science).

## Silver staining

Samples from the purification were loaded on 12% polyacrylamide gels (Bio-Rad) and run for 60 minutes at 150 volts. The gels were then incubated for 30 minutes at room temperature in fixing solution (50% methanol, 12% acetic acid) and afterwards incubated in SDS removal solution (10% ethanol, 5% acetic acid) for 30 minutes at room temperature. Next, the gels were transferred to Farmer's solution to enhance PrP staining (containing: 0.15 g potassium ferricyanide, 0.3 g sodium thiosulfate, 0.05 g sodium carbonate) for 2 minutes, and then washed three times in distilled water. Next, the gels were treated with 0.2% (w/v) AgNO$_3$ for 20 minutes and rinsed briefly in distilled water, followed by 100 ml developing solution (15 g sodium carbonate and 250 microL 30% formaldehyde stock solution). Finally, the development stopped in a solution of 0.2% acetic acid [37].

## Negative stain electron microscopy

10-μl drops of purified samples were adsorbed onto freshly glow-discharged 400 mesh carbon-coated copper grids (Electron Microscopy Sciences) for 1 minute and washed in 1–3 drops of (50 μl) 0.1 M and 0.01 M ammonium acetate solutions each. Then, the grids were stained using a freshly filtered 2% solution of uranyl acetate and air-dried after removing the excess stain with filter paper. The stained samples were examined with a Tecnai G20 transmission electron microscope (FEI Company) operating at an acceleration voltage of 200 kV. Electron micrographs were recorded with an Eagle 4k x 4k CCD camera (FEI Company). Defocus levels of 1–2 μm were applied for recording the images. Micrographs acquired for image processing purposes, were recorded at a nominal magnification of 29,000x with a pixel size of 3.07 Å per pixel.

## Width measurements

Width measurements were performed on L-type BSE fibrils using EMAN's boxer [51]. For each fibril, the maximum diameter between two crossover regions was determined.

## Immunogold labeling

Immunogold labeling of the L-type BSE fibrils was performed using a combination of anti-PrP antibodies, including Fab fragments Fab 69 and Fab 29, and monoclonal antibody YEG mAb Sc-G1, detecting prion protein epitopes at different positions. The Fab fragments were selected from a phage display library. Fab 69 and Fab 29 react with epitopes within residues $_{100}$GWG QGGTHGQW$_{110}$ and $_{227}$TQYQRESQAYY$_{237}$ located at N- and C-terminal regions of the bovine prion protein, respectively [52]. The Fabs were expressed in *E.coli* (BL21(DE3)) as a His-tagged, soluble, and functional protein. The Fab fragments were purified via IMAC chromatography, following lysis of the bacterial cells. Next, the fragments were eluted using imidazole buffer, desalted, and further assessed for purity and activity. The YEG mAb Sc-G1 monoclonal antibody was produced in-house and recognizes a discontinuous epitope in native PrP$^{Sc}$ only.

Based on a previously published immunogold labeling protocol [53], 7 μl of purified L-type BSE samples were adsorbed onto glow discharged formvar/carbon-coated nickel grids (Ted Pella, Inc.) for ~5 minutes, and washed using three drops (50 μl) of 0.1 M and 0.01 M ammonium acetate buffer pH 7.4. Samples used for labeling with Fab 69 and Fab 29 antibodies, were treated with 50 μl of 3 M urea for 10 minutes, to increase the epitope accessibility. Following the washing steps, the grids were stained with 2 drops of freshly filtered 2% sodium phosphotungstic acid (PTA), pH 7.2, then blocked for 60 minutes with 0.3% bovine serum albumin (BSA) in Tris buffered saline (TBS: 50 mM Tris-HCl, pH 7.4; 150 mM NaCl). Next, all grids, except for the control samples, were incubated on 50 μl drops of primary antibodies YEG mAb Sc-G1, Fab 69, and Fab 29 for 2.5–3 hours, followed by 5 washes in 1% BSA. The grids that were treated with Fab 69 and Fab 29 were transferred onto droplets of goat F(ab')2 anti-human IgG F(ab')2 (Abcam ab98531) for 2 hours, followed by washing steps, incubated with a 5-nm-gold-conjugated rabbit anti-goat IgG (Abcam ab202670) for 2 hours and rinsed 5 times in 0.1% BSA. For the YEG mAb Sc-G1 antibody, incubation with the primary antibody and washes in 1% BSA, followed by treatment with 50 μl droplets of the gold-conjugated secondary antibody, 6 nm goat anti-mouse IgG (Abcam ab39614), diluted 1/50 in blocking buffer, for 2.5 hours and 5 washes in 0.1% BSA. Finally, the grids were rinsed with TBS solution and water, and placed onto two drops of 2% PTA for final staining, air-dried, and stored for EM analysis. The control experiments were conducted similarly except for the omission of the primary antibodies. The samples were analyzed with a Tecnai G20 transmission electron microscope (FEI

Company) operating at an acceleration voltage of 200 kV. Electron micrographs were recorded with an Eagle 4k x 4k CCD camera (FEI Company).

## 2D class averaging

Two-dimensional classification is a computational tool performed on electron micrographs to enhance the signal-to-noise ratio (SNR) of the images. This involves the alignment and grouping of the dataset into classes based on morphological similarities [44]. Reference-free 2D class averaging was performed by applying boxes with a size of 200 by 200 pixels and a 50% overlap along the fibrils' axis between two crossovers using EMAN's boxer program [51]. Classification and averaging were implemented on a total of 532 segmented particles of one-protofilament fibrils and 297 segmented particles of two-protofilament fibrils for a total of n = 3 classes.

## 3D helical reconstruction

For helical reconstructions, we selected representative isolated fibrils with an apparent helical twist from among hundreds of electron micrographs, all previously examined visually for the width measurements. For this method, individual fibrils were segmented along the fibril axis covering at least two crossovers into overlapping boxes with the size of 300 by 300 pixels, equal to 92.1 by 92.1 nm, and 95% to 99% overlap between adjacent segments using EMAN's boxer software [24,51]. All segments were aligned, and the angular orientation of each box was calculated based on the fibril's repeat distance, followed by generation of a preliminary 3D map by back-projection of all boxed 2D projections using the image processing software SPIDER [54]. Next, the reconstruction was low-pass filtered to 20 Å, and the corresponding symmetry was imposed (no symmetry for the one-protofilament fibrils and two-fold symmetry for the two-protofilament fibrils). Afterwards, the initial model underwent further refinements, including alignment of the reconstruction to other reconstructions of the same fibril with different segmentation overlaps [55], which continued by averaging aligned 3D volumes through an iterative process of alignment and averaging.

Extended helical reconstruction: This approach was implemented on the final reconstructions in order to eliminate the rippling artifacts from the overlapping boxes [24]. Briefly, the refined reconstructed volume was sliced across the yz plane, following 90˚ rotation around its x-axis direction, which yielded a set of 2D projections of the fibril's cross-section. Then, the stack of 2D projections underwent alignment and averaging processes iteratively. Next, the IMAGIC software was used to center and replicate the final 2D averaged image [56]. Finally, rotation angles were applied to the set of replicated averaged 2D cross-sections according to the helical repeat, and a 3D map was built using IMAGIC software [56]. All 3D reconstructions were visualized in UCSF Chimera [57].

## Supporting information

**S1 Fig. Two representative EM micrographs of semi-purified pellet 1 L-type BSE samples.** The pellet 1 sample was taken before the sucrose step gradient ultracentrifugation. (TIF)

## Acknowledgments

We would like to thank Dr. Stanley B. Prusiner (Department of Neurology, University of California, San Francisco, California, USA) for kindly providing the Tg4092 mice and Dr. Stefanie Czub (Canadian BSE Reference Laboratory, Canadian Food Inspection Agency, Lethbridge Laboratory, Lethbridge, Alberta, Canada) for generously providing the original L-type BSE

prion samples. Furthermore, we thank Geraldine Horny, Sylvie Fels, Nathalie George, Stefan Ewert, and Thomas Pietzonka from the Novartis Institutes for BioMedical Research (NIBR Basel, Switzerland) for providing the antibody phage display library from which the anti-PrP Fab 29 and Fab 69 originate.

## Author Contributions

**Conceptualization:** Holger Wille.

**Data curation:** Razieh Kamali-Jamil, Holger Wille.

**Formal analysis:** Razieh Kamali-Jamil, Ester Vázquez-Fernández, Howard S. Young, Holger Wille.

**Funding acquisition:** Holger Wille.

**Investigation:** Razieh Kamali-Jamil, Brian Tancowny, Vineet Rathod, Xiongyao Wang, Sandor Dudas.

**Methodology:** Howard S. Young, Holger Wille.

**Project administration:** Holger Wille.

**Resources:** Vineet Rathod, Xinli Tang, Andrew Fang, Assunta Senatore, Simone Hornemann, Sandor Dudas, Adriano Aguzzi.

**Software:** Ester Vázquez-Fernández, Sara Amidian, Howard S. Young.

**Supervision:** Holger Wille.

**Validation:** Holger Wille.

**Visualization:** Razieh Kamali-Jamil, Ester Vázquez-Fernández, Xiongyao Wang, Howard S. Young.

**Writing – original draft:** Razieh Kamali-Jamil, Holger Wille.

**Writing – review & editing:** Razieh Kamali-Jamil, Holger Wille.

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
