## [Decision Letter · Decision Letter 0]

10 Mar 2021

Dear Dr. Wille,

Thank you very much for submitting your manuscript "The ultrastructure of infectious L-type bovine spongiform encephalopathy prions constrains molecular models" for consideration at PLOS Pathogens. As with all papers reviewed by the journal, your manuscript was reviewed by members of the editorial board and by several independent reviewers. In light of the reviews (below this email), we would like to invite the resubmission of a significantly-revised version that takes into account the reviewers' comments.

We cannot make any decision about publication until we have seen the revised manuscript and your response to the reviewers' comments. Your revised manuscript is also likely to be sent to reviewers for further evaluation.

Sincerely,

Surachai Supattapone

Associate Editor

PLOS Pathogens

Neil Mabbott

Section Editor

PLOS Pathogens

Kasturi Haldar

Editor-in-Chief

PLOS Pathogens

orcid.org/0000-0001-5065-158X

Michael Malim

Editor-in-Chief

PLOS Pathogens

orcid.org/0000-0002-7699-2064

Reviewer's Responses to Questions

**Part I - Summary**

Reviewer #1: This intriguing study from a foremost group in this area provides EM-based morphological analyses of L-BSE prions produced in bovinized mice. Interestingly, two fibril morphologies were seen, containing either one or two protofilaments, with the one-protofilament fibrils being more common. This work should make a valuable contribution to the field. However, I have concerns that should be addressed about the characterization of the fibril preparations as well as the authors’ interpretations of their data.

Reviewer #2: This is a qualitative EM study by Kamali-Jamil et al., to describe the ultrastructure of BSE prions derived from transgenic mouse models. The authors observe interesting single and double protofilament morphologies, and use micrographs of such for further 2D class averages and 3D reconstructions. Based on this, they derive low resolution ultrastructural models which they argue are most compatible with 4RbS molecular models for prion structure.

The EM data is certainly interesting, as are the single and double protofilament arrangements of the rodent-passaged BSE prions. However, the main point made seems to be that this data only supports a 4RbS model, and this is not sufficiently supported by the presented data. The resolution of the reconstructions is insufficient to conclude the fibrils are formed by 4RbS (or any other model). This study also relies on negatively stained fibrils, which helps with visualization of the fibrils, but could conceivably introduce artifacts into the imaging data and subsequent reconstruction analyses related to the expected variabilities of negative stain densities.

The observation of one versus two protofilaments is used as direct evidence to argue for a 4RbS molecular model and most directly against PIRIBS. The authors don't address that there are many examples of two protofilament fibrils that are formed from two parallel, in register architectures including examples for tau, aSyn, and prion protein fragments. Further, the cross-section of a single 4RbS looks to be much smaller than the density observed in the single BSE protofilament. If a 4RbS, one might even expect multiple 4bS within the single protofilament width described here, especially as it's unclear if glycans and lipid anchors would be resolved after 2D class averages and 3D reconstructions. Neither the reconstructed topologies, or the Fab data (without further biochemical evidence of what amino acid/structural elements are found at the interface between the two protofilaments) convincingly support or exclude any molecular model (4RbS, PIRIBS, or otherwise).

Very recently posted descriptions suggest that at least one mammalian prion is a piribs (with fibrils of comparable widths to those noted here for the single BSE protofilament). The authors might consider if alternative models to a single 4RbS per filament are also compatible within their EM data.

Reviewer #3: The manuscript describes structural studies of a specific L-type strain of infectious prions. Structural studies of brain-derived prions are extremely complicated, so our knowledge on the structure of infectious prions is still limited and each new piece of information is important. So, I strongly support publication of this paper in PLoS Pathogens.

The experiments are well-executed, and the paper is well-written, so I have just several concerns

**Part II – Major Issues: Key Experiments Required for Acceptance**

Reviewer #1: Fig. 1: The staining in the gradient fraction lanes is too weak to be useful, i.e., almost invisible. How do these purified bands compare to L-BSE fibrils prepared from cattle brain? Side-by-side lanes should be provided for bovine- and mouse-derived preps, if possible. At the very least, appropriate comparisons to gel data for L-BSE in the literature should be made. What, if anything, has happened to the dark silver-stained band in the "final pellet in panel A, and "pellet 1" in panel C, on the subsequent sucrose gradient? Most importantly, can the authors provide evidence for or against the possibility that with passage of L-BSE into bovinized mice that another strain that might have arisen that accounts for either the single or double protofilament ultrastructures in their preparations? Such prion strain alterations are more likely to occur during interspecies transmissions.

Reviewer #2: (No Response)

Reviewer #3: 1. The only concern which may be considered as major is explanation of the high-intensity band in the silver-stained gel (Figure 1A, line “Final Pellet”). When I compare it to the western blot gel (Fig. 1B), it’s quite clear that the top band in “Final Pellet” should correspond to di-glycosylated PrP, which is known as PrP27-30 in literature, a bit lower weak band is for mono-glycosylated PrP, while the third band should be non-glycosylated PrP. But there is a major band at the very bottom of the gel, which has no corresponding band in the western blot. If it can’t be explained as some kind of artifact, then identification of what protein or peptide corresponds to this band is crucial for the whole study, as this protein/peptide seems to be the major component of the final pellet.

**Part III – Minor Issues: Editorial and Data Presentation Modifications**

Reviewer #1: 1. The authors state in multiple places, such as line 201-2, that single protofilament prion fibrils have not been reported previously, but this is not true. See, for example, Sim & Caughey, Neurobiol. Aging 2009. Thus, the authors need to remove such claims of novelty.

2. L218-219: “Moreover, the presence of two distinct and separable protofilaments in these images contradicts the PIRIBS model…” This is not true because two high resolution cryo-EM structures of synthetic PrP fibrils with PIRIBS architectures feature two protofilaments. See C. Glynn et al., Nat. Struct. Mol. Biol., 2020 & L. Q. Wang et al., Nat Struct Mol Biol 2020.

3. L221-3: “These micrographs preclude the possibility that stain artifacts contribute to the observed gap between the protofilaments, as the protofilaments separate into distinct fibrils.” This statement should be softened because such results could also be obtained if PK cleaved an exposed chain that links two domains of a PIRIBS-based fibril. For unequivocal evidence that distinct N- and C-terminal domains can exist in infectious prion fibrils, see https://www.biorxiv.org/content/10.1101/2021.02.14.431014v1.

4. Indeed, although the authors would not have known about this recently posted high-resolution prion structure in bioRxiv prior to submission of their manuscript, I would encourage them to adjust their interpretations and conclusions during revision with the bioRxiv data in mind (see below). I would argue that broadening their conclusions to include the possibility of PIRIBS-based architectures for L-BSE fibrils will improve, rather than detract from, the strength of their findings.

5. L290-2: “Superimposition of the recent 4RβS model combined with N-linked glycans (24, 25) with cross-sections from our 3D reconstructions showed that these reconstructions fit well with the 4RβS model (Fig 10).” Certainly the 4RβS model fits within the cross-section, but it seems to be much too small to fill the space.

6. L376-7: “An abundance of experimental evidence indicates that the core structure of the infectious prion contains a 4RßS fold”. This is stated too strongly, especially given the recent posting of a high-resolution prion structure with a PIRIBS architecture (see bioRxiv link above). I would suggest that “indicates” be replaced with “is compatible with”.

7. L385-387: “From the two main models for prion structure, our findings are compatible with the ß-solenoid model only as we identified both one- and two-protofilament fibrils in our samples of purified L-type BSE prions”. I would strongly caution against such a categorical statement. First, as referenced above, PrP fibrils with PIRIBS architectures have been shown with high resolution to be able to have one- or two-protofilaments. Second, the authors seem to be implying that the size of their single-protofilament L-BSE fibril cores (i.e. averaging 10.6 nm) are too small to be compatible with a PIRIBS architecture. For example, in the Abstract they conclude that “The fact that the one protofilament fibrils contain both N- and C-terminal PrP epitopes constrains molecular models for the structure of the infectious conformer in favour of a compact four-rung β-solenoid fold and excludes more extended fold models.” However, the ordered PIRIBS-based core of 263K scrapie prion fibrils is ~4 x 13 nm (see bioRxiv link above), and one can readily imagine that plausible strain-dependent permutations in this structure could reduce the largest dimension of the core by a mere 1.4 nm, while still maintaining an extended chain PIRIBS-based architecture.

Reviewer #2: 1) Figure 1 figures are cropped very close to the bands of interest. The authors should show an expanded image of the blots/gels.

2) How are the crossover distances being defined and determined for the two protofilament fibrils? At least in the micrographs shown it seems sometimes variable - an example micrograph with crossovers indicated would be helpful.

3) In Methods, pixels are used to describe the box size used in particle selection and reconstructions. What is the Å/pixel conversion? Imaging magnification?

4) The immunogold labelling appears to also label amorphous material in addition to the fibrils. Is this a frequent observation?

Reviewer #3: 2. The authors state “Moreover, the presence of two distinct and separable protofilaments in these images contradicts the PIRIBS model, which proposed an in-register stacking of PrPSc monomers covering the full width of a “two protofilament” structure” and “From the two main models for prion structure, our findings are compatible with the β-solenoid model only, as we identified both one- and two-protofilament fibrils in our samples of purified L-type BSE prions”. I can’t agree that these statements are valid, as the one-protofilament L-type BSE fibril has similar thickness as two-protofilament fibril, which was used to propose PIRIBS model. From the studies of other proteins, which can form amyloid fibrils we know that morphology of fibrils may change with time (Adamcik, J., Jung, J.-M., Flakowski, J., De Los Rios, P., Dietler, G., and Mezzenga, R. (2010). Understanding amyloid aggregation by statistical analysis of atomic force microscopy images. Nature Nanotechnology 5, 423–428. ; Adamcik, J., Castelletto, V., Bolisetty, S., Hamley, I.W., and Mezzenga, R. (2011). Direct observation of time-resolved polymorphic states in the self-assembly of end-capped heptapeptides. Angewandte Chemie - International Edition 50, 5495–5498. ; Usov, I., Adamcik, J., and Mezzenga, R. (2013). Polymorphism Complexity and Handedness Inversion in Serum Albumin Amyloid Fibrils. ACS Nano 7, 10465–10474.,) for example thin protofilaments, twisted together, after longer incubation may look like one thicker filament. I think that brain-derived material consists mostly of mature fibrils, so one-protofilament fibril may in fact contain two protofilaments, just we have not enough resolution to detect it.

3. Finally, I think the statement in results section “The mass difference between these two prion variants can easily explain the apparent difference fibril widths” contradicts with the information in discussion section “The maximum fibril width for bovine peptide fibrils was found to be 17.0 nm (49), which is close to the width of our two-protofilaments fibrils of bovine PrP 27-30 prions.”

PLOS authors have the option to publish the peer review history of their article (what does this mean?). If published, this will include your full peer review and any attached files.

Reviewer #1: **Yes: **Byron Caughey

Reviewer #2: No

Reviewer #3: **Yes: **Vytautas Smirnovas
---

## [Decision Letter · Decision Letter 1]

10 May 2021

Dear Dr. Wille,

We are pleased to inform you that your manuscript 'The ultrastructure of infectious L-type bovine spongiform encephalopathy prions constrains molecular models' has been provisionally accepted for publication in PLOS Pathogens.

Best regards,

Surachai Supattapone

Associate Editor

PLOS Pathogens

Neil Mabbott

Section Editor

PLOS Pathogens

Kasturi Haldar

Editor-in-Chief

PLOS Pathogens

orcid.org/0000-0001-5065-158X

Michael Malim

Editor-in-Chief

PLOS Pathogens

orcid.org/0000-0002-7699-2064

Reviewer Comments (if any, and for reference):

Reviewer's Responses to Questions

**Part I - Summary**

Reviewer #1: the authors have adequately addressed my concerns.

Reviewer #2: The authors have improved the manuscript with additional considerations of alternate models and inclusion of labelling and additional technical details to satisfactorily address my major concerns. I have only two minor editorial/data presentation clarifications to note.

Reviewer #3: The authors have adequately responded to my comments and made necessary corrections, so I recommend to accept the paper for publication.

**Part II – Major Issues: Key Experiments Required for Acceptance**

Reviewer #1: none

Reviewer #2: (No Response)

Reviewer #3: (No Response)

**Part III – Minor Issues: Editorial and Data Presentation Modifications**

Reviewer #1: none

Reviewer #2: Figure 1 (particularly Fig 1B) include MW marker labels to indicate sizes below 25 kDa

Line 256 - Without direct experimental evidence indicating the thicker fibrils are indeed less prone to breakage, I would suggest using the term “suggests” or equivalent instead of “appears” (i.e. suggests a reduced sensitivity . . .)

Reviewer #3: (No Response)

PLOS authors have the option to publish the peer review history of their article (what does this mean?). If published, this will include your full peer review and any attached files.

Reviewer #1: **Yes: **Byron Caughey

Reviewer #2: No

Reviewer #3: **Yes: **Vytautas Smirnovas

---

## [Editor Report · Acceptance letter]

28 May 2021

Dear Dr. Wille,

We are delighted to inform you that your manuscript, "The ultrastructure of infectious L-type bovine spongiform encephalopathy prions constrains molecular models," has been formally accepted for publication in PLOS Pathogens.

Best regards,

Kasturi Haldar

Editor-in-Chief

PLOS Pathogens

orcid.org/0000-0001-5065-158X

Michael Malim

Editor-in-Chief

PLOS Pathogens

orcid.org/0000-0002-7699-2064